

# ATTILA 4.0: Lagrangian Advective and Convective Transport of Passive Tracers within the ECHAM5/MESSy (2.53.0) Chemistry Climate Model

Sabine Brinkop[1,2] and Patrick Jöckel[1]

[1]Deutsches Zentrum für Luft- und Raumfahrt, Institut für Physik der Atmosphäre, Oberpfaffenhofen, 82230 Wessling, Germany
[2]Meteorologisches Institut der Universität München, 80333 München, Germany

**Correspondence:** S. Brinkop (Sabine.Brinkop@dlr.de)

**Abstract.** We have extended ATTILA (Atmospheric Tracer Transport in a LAgrangian model), a Lagrangian tracer transport scheme, which is on-line coupled to the global ECHAM/MESSy Atmospheric Chemistry (EMAC) Climate model, with a combination of newly developed and modified physical routines, and new diagnostic and infrastructure submodels. The new physical routines comprise a parametrisation for Lagrangian convection, a formulation of diabatic vertical velocity, and the new grid-point submodel LGTMIX to calculate the mixing of compounds in Lagrangian representation. The new infrastructure routines simplify the transformation between grid-point (GP) and Lagrangian (LG) space in a parallel computing environment. The new submodel LGVFLUX is a useful diagnostic tool to calculate on-line vertical mass-fluxes through horizontal surfaces. The submodel DRADON was extended to account for emissions and changes of $^{222}$Rn on Lagrangian parcels. To evaluate the new physical routines, two simulations in free-running mode with prescribed sea surface temperatures were performed with EMAC-ATTILA in T42L47MA resolution from 1950 to 2010. The results show an improvement of the tracer transport into and within the stratosphere, when the diabatic vertical velocity is used for vertical advection in ATTILA instead of the standard kinematic vertical velocity. Especially the age-of-air distribution is more in accordance with observations. The global tropospheric distribution of $^{222}$Rn, however, is simulated in agreement with available observations and with the results from EMAC in grid-space for both Lagrangian systems. Additional sensitivity studies reveal an effect of the inter-parcel mixing on the age-of-air in the tropopause region and the stratosphere, but no significant effect for the troposphere.

## 1 Introduction

Due to the increasing demand for including interactive tracers in climate simulations it becomes necessary to use global models which meet the needs of a fast and exact tracer transport scheme. Commonly used methods to describe the large-scale transport in a general circulation model of the atmosphere follow the Eulerian method. The Lagrangian (LG) method (i.e., from the perspective of a fluid particle or parcel) is more frequently used off-line for trajectory studies in particle models like in the global 3-d chemistry transport model (Collins et al., 2002), in FLEXPART (Stohl et al., 1998) or CLaMs (McKenna et al., 2002). An exception are the LG models ATTILA (Atmospheric Tracer Transport in a LAgrangian model, Reithmeier



and Sausen (2002)), and CLaMs, which has recently been coupled to the global chemistry-climate model EMAC (Hoppe et al., 2014). Describing the transport of tracers by a LG transport scheme has advantages compared to an Eulerian transport method: mass conservation (not in CLaMS) and absence of numerical diffusion. These advantages become most important, if tracer distributions are inhomogeneous with strong vertical or horizontal gradients (Stenke and Grewe, 2005), which ought

to be smoothed by physical and not by numerical diffusion processes. In this study we introduce the extended and improved LG advection scheme ATTILA, being parallelised, modularised and rewritten as a submodel for EMAC (Jöckel et al., 2010). ATTILA was originally developed by Reithmeier and Sausen (2002). We implemented a LG convection scheme and a diabatic vertical velocity formulation, which can be selected instead of the standard kinematic vertical velocity. The need of these two physical improvements is due to the following reasons:

First, the large-scale transport of trace species is sensitive to the selected vertical velocity scheme (Eluszkiewicz et al., 2000). Therefore, several studies recommend the use of a diabatic vertical velocity for the representation of LG transport in the tropical tropopause layer and the stratosphere (Eluszkiewicz et al., 2000; Ploeger et al., 2010; Hoppe et al., 2016). Specific transport characteristics like the residence time in the tropical tropopause layer (TTL) and the pathways to the stratospheric over-world are simulated more realistically (Ploeger et al., 2010; Hoppe et al., 2014, 2016) with a diabatic vertical velocity.

Typically, kinematic velocities are calculated as a residual from the horizontal flux divergence using the continuity equation. Because horizontal velocities are two orders of magnitude larger than the vertical velocity, kinematic velocities show up rather noisy.

Second, convective transport is an important fast vertical transport process for trace species in the troposphere and tracer distributions are sensitive to the convection parametrisation (Mahowald et al., 1995; Tost et al., 2006; Erukimova and Bowman,

2006; Zhang et al., 2008). The vertical tracer distribution depends on the accuracy of transport from the boundary layer, where the chemical species are emitted, into the free troposphere. Two LG transport schemes are known to use a convection parametrisation: The CLaMS transport model considers convection by using the moist Brunt-Väisälä frequency parametrisation to include the effects of vertical instability on the related convection (Tao, 2016; Konopka et al., 2018). In the FLEXPART transport model (Stohl et al., 1998) the convection scheme relies on the ECMWF grid-scale temperature and humidity and

provides a matrix for the vertical convective particle displacement (Seibert et al., 2002; Forster et al., 2007).

In the former (non-parallelised) version of ATTILA, convective tracer tendencies were calculated in grid-point space and then transformed onto the parcels (Reithmeier and Sausen, 2002). This transformation, however, is not mass-conserving. Moreover, parcel trajectories do not follow convective up- and downdrafts. This is a drawback with respect to the analysis of trajectories, which were subject to convective uplift, and the motivation to incorporate a LG convection scheme in ATTILA. Besides the

transport of parcels, mixing of compounds between adjacent parcels is an important process, that reduces gradients of trace gases horizontally and vertically. Physically, the character of turbulence in the atmosphere (due to wind shear or buoyancy) controls the degree of mixing. A LG model, that successfully uses a physical parametrisation for mixing based on the atmospheric flow deformation is ClaMS (McKenna et al., 2002; Konopka et al., 2004; Riese et. al,, 2012). However, our parametrisation of mixing, realised in the new submodel LGTMIX, is so far only based on two parameters: one for the troposphere and one for

the stratosphere, respectively, and represents local isotropic turbulent mixing. This concept was already successfully applied by



Reithmeier and Sausen (2002). However, LGTMIX is written to allow for the incorporation of more physically sound mixing parametrisations more easily in the future.

In Sect. 2 we shortly repeat the main concepts of ATTILA, which were published in detail by Reithmeier and Sausen (2002), and we introduce the application and extensions of the MESSy infrastructure, and the concept of the calculation
of random numbers in a parallel computing environment. Additionally, we describe the new LG convection parametrisation and the diabatic velocity of the new ATTILA version. The turbulent mixing of compounds between the parcels (submodel LGTMIX), the extended diagnostics (submodel LGVFLUX), the extentions to the submodel DRADON to handle $^{222}$Rn in LG representation, and the submodel LGGP, calculating the transformations between GP and LG representation, are also described in Sect. 2. The observational data for comparison are described in Sect. 3. Sect. 4 describes the model simulations performed
with ATTILA coupled to the global chemistry climate model EMAC. The evaluation of the LG simulations is presented in Sect. 5. We compare the LG simulation results with observations and also with EMAC (GP) simulations, which were already evaluated by Jöckel et al. (2016).

## 2   Model description

### 2.1   EMAC - a MESSy-fied global chemistry-climate model

The ECHAM/MESSy Atmospheric Chemistry (EMAC) model is a numerical chemistry and climate simulation system that includes submodels describing tropospheric and middle atmosphere processes and their interaction with oceans, land and human influences (Jöckel et al., 2010). It uses the second version of the Modular Earth Submodel System (MESSy2) to link multi-institutional computer codes. The core atmospheric model is the 5th generation European Centre Hamburg general circulation model (ECHAM5 Roeckner et al., 2006). For the present study we applied EMAC (ECHAM5 version 5.3.02,
MESSy version 2.53.0 in the T42L47MA-resolution, i.e. with a spherical truncation of T42 (corresponding to a quadratic Gaussian grid of approx. 2.8°by 2.8°in latitude and longitude) with 47 vertical hybrid pressure levels up to 0.01 hPa (MA-middle atmosphere). The applied model setup comprised the submodels listed in Table 1.



**Table 1.** List of MESSy submodels used for the simulations in this study.

| submodel | description | reference(s) |
|---|---|---|
| AEROPT | AERosol OPTical properties | Dietmüller et al. (2016) |
| ATTILA | Atmospheric Tracer Transport In a LAgrangian model | Reithmeier and Sausen (2002); Sect. 2.2 |
| CH4 | methane oxidation and feedback to stratospheric water vapour | |
| CLOUD | ECHAM5 cloud scheme as MESSy submodel | Roeckner et al. (2006, and references therein) |
| CLOUDOPT | cloud optical properties | Dietmüller et al. (2016) |
| CONVECT | convection parameterisations | Tost et al. (2006) |
| CVTRANS | convective tracer transport | Tost (2006) |
| DRADON | $^{222}$Rn and decay products as diagnostic tracers | Jöckel et al. (2010) |
| E5DIFF | ECHAM5 vertical diffusion scheme as MESSy submodel | Roeckner et al. (2006, and references therein) |
| GWAVE | ECHAM5 gravity wave parametrisation as MESSy submodel | Roeckner et al. (2006, and references therein) |
| JVAL | photolysis rates | Sander et al. (2014) |
| LGGP | transformation between LG and GP and vice versa | Sect. 2.3 |
| LGTMIX | LaGrangian Tracer MIXing | Sect. 2.4 |
| LGVFLUX | LaGrangian based Vertical FLUX analyses | Sect. 2.5 |
| OFFEMIS[1] | OFFline (i.e., prescribed) EMISsions of tracers | Kerkweg et al. (2006) |
| ORBIT | Earth ORBITal parameters as MESSy submodel | Roeckner et al. (2006, and references therein) |
| OROGW | ECHAM5 OROgraphic gravity wave parameterisation as MESSy submodel | Roeckner et al. (2006, and references therein) |
| PTRAC | Prognostic TRACers defined via namelist | Jöckel et al. (2008) |
| QBO | Newtonian relaxation of quasi-biennial oscillation | Giogetta and Bengtson (1999); Jöckel et al. (2006) |
| RAD | ECHAM5 radiation scheme as MESSy submodel | Dietmüller et al. (2016) |
| RAD_FUBRAD | high-resolution short-wave radiation sub-submodel | Nissen et al. (2007); Dietmüller et al. (2016) |
| SURFACE | ECHAM5 surface scheme as MESSy submodel | Roeckner et al. (2006, and references therein) |
| TNUDGE | Newtonian relaxation of tracers as pseudo-emissions | Kerkweg et al. (2006) |
| TROPOP | tropopause and other diagnostics | Jöckel et al. (2006) |
| VAXTRA | Vertical AXes TRansformations (for output) | |
| VISO | iso-surfaces and maps | Jöckel et al. (2010) |

[1] formerly named
OFFLEM

The following list gives an overview over the modified and newly developed routines, presented in more detail in the following sections:

a. Modifications and extensions of physical processes:

– Additional subroutines for ATTILA to describe Lagrangian convection have been added.



– A formulation of vertical movement of air parcels in ATTILA based on the diabatic vertical velocity has been implemented.

– A new submodel (LGTMIX) to calculate the mixing of compounds in Lagrangian representation has been implemented.

– The submodel DRADON to account for emission and decay of $^{222}$Rn has been expanded for the new Lagrangian representation of tracers.

b. New diagnostic and infrastructure submodels:

– A new submodel for the infrastructure, such as for the calculation of random numbers in a parallel environment has been implemented.

– A sub-submodel that calculates the transformations needed in ATTILA (ATTILA_TOOLS) has been added.

– A new submodel that calculates the transposition of variables between Lagrangian and grid point space and vice versa (LGGP) for the output has been implemented.

– A new submodel to diagnose the vertical fluxes through horizontal surfaces (LGVFLUX) has been implemented.

## 2.2 Submodel ATTILA: Atmospheric Tracer Transport In a LAgrangian model

ATTILA is a Lagrangian tracer transport scheme, now including LG convection, which can optionally be selected to transport tracers in Lagrangian representation, in addition to the standard flux-form semi-Lagrangian (FFSL) scheme (Lin and Rood, 1996) for tracers in GP representation. ATTILA runs on-line as submodel within EMAC. A former version of ATTILA has been described in detail by Reithmeier and Sausen (2002). The main concepts of ATTILA are shortly repeated in this section (time-stepping procedure, interpolation methods, initialisation) and complemented by new infrastructure (random number generator, parallelisation, transformation and transposition methods), new physical (air parcel mixing, Lagrangian convection) and new diagnostic submodels.

In ATTILA the atmospheric mass is divided into single mass packets, which have an equal air mass loading but no volume. The parcels are regarded as centroids, when they are advected with the wind field provided by the spectral dynamical core of EMAC. The number of parcels within the atmosphere is only limited by the available computational resources. A typical choice is an average of 3 parcels per EMAC grid box, similar as documented by Reithmeier and Sausen (2002). However, the actual number of parcels per grid box may vary between zero and 10, depending on the vertical and horizontal size of the grid box.

### 2.2.1 Model infrastructure

To enable ATTILA in a distributed memory parallel environment (e.g., applying the Message Passing Interface, MPI) we chose to follow a domain cloning approach. Whereas the base-model EMAC follows a classical horizontal domain decomposition





approach for distributed memory parallelisation, we distribute the global number $N$ of ATTILA air parcels, which keep their identity throughout a simulation, (almost) equally among the parallel tasks (index $i$),

$$N = \sum_{i=0}^{p-1} n_i \ ,$$ (1)

where $p$ is the number of parallel tasks and $n_i$ the number of parcels bound to task $i$. Note that $n_i = n_j$ for all $(i,j)$, except for

$i = p - 1$, depending weather $N$ is divisible by $p$ or not.

During the simulation, each parcel keeps being bound to its initial task. Since all parcels on each task move around the entire globe with time, it is necessary to provide the required input variables to drive ATTILA (such as for instance the wind velocity vector from EMAC) as global fields (i.e., by cloning of the global domain of these variables). The subroutines for data transpositions between parallel decomposed grid-point and corresponding cloned global variables have been added to the

MESSy infrastructure submodel TRANSFORM.

To facilitate the exchange of Lagrangian objects between Lagrangian enabled submodels as so-called *channel objects* (see Jöckel et al. (2010) for a detailed explanation of the MESSy infrastructure submodel CHANNEL), we define a new *representation* (see Jöckel et al. (2010) for a detailed explanation) of rank 1, global dimension length $N$, and local (i.e. task-specific) dimension length $n_i$. The corresponding MPI-based gather- and scatter-routines for serial netcdf I/O have been added to the

MESSy infrastructure submodel TRANSFORM. This new *representation*, named LG_ATTILA, is used by the Lagrangian submodels to define their specific Lagrangian objects.

For tracers we further define two additional *tracer sets* (see Jöckel et al. (2008) for a detailed explanation) one ("tracer_lg") in the new Lagrangian *representation* to handle the Lagrangian tracers, and one "tracer_lggp" in grid-point representation. The latter is solely used to transform the Lagrangian tracers into grid-point space for output and further analyses.

Subroutines to transform and transpose variables between Lagrangian representation and (parallel decomposed) grid-point representation, and vice versa, are collected in a specific tool-box module named ATTILA_TOOLS. This comprises also specific subroutines for the transformation of grid-point emission fluxes into Lagrangian tracer tendencies. For the latter, four options are implemented: The emitted mass from a grid cell is distributed

1. evenly among all LG parcels in that grid box. In case there is no parcel at a given time in that grid box, the mass is stored
and accumulated over time, and eventually released into the next parcel(s) passing by.

2. evenly among all LG parcels in the lowest grid-box of the boundary layer with at least one parcel in it. In case the entire column in the boundary layer is empty (i.e., no parcels) at a given time, the mass is stored, accumulated and eventually released to the next parcel as in 1.

3. among all parcels in the boundary layer, however, weighted with a linear, negative vertical gradient. The treatment of
empty boundary layer columns as in 1. and 2.

4. evenly among all parcels in the boundary layer. The treatment of empty boundary layer columns as in 1., 2., and 3.





ATTILA requires up to four series of pseudo-random numbers, one for the boundary layer turbulence parametrisation (Sect. 2.2.3), one for the convection parametrisation (Secttion 2.2.4), one for an envisaged (but not yet implemented) additional clear air turbulence parametrisation, and one for particle displacements parametrising a Monte Carlo diffusion approach. One additional pseudo-random number series is used for the initial distribution of the parcels in the model atmosphere. These

pseudo-random number series are provided by the MESSy infrastructure submodel RND. RND provides uniformly distributed pseudo-random numbers between 0 and 1, calculated with either the standard Fortran90 function, RANDOM_NUMBER, or the Mersenne Twister algorithm (Matsumoto and Nishimura, 1998), or the Luxury algorithm (Lüscher, 1994; James, 1994). Based on these, RND can also provide normally-distributed random numbers centred around zero, using the Marsaglia polar method[2]. The generation of high-quality pseudo-random number series in a parallel environment is not straightforward. Seeding inde-

pendent series on each task implies the high risk that these series become correlated. Moreover, the result is decomposition dependent, i.e. it depends on the number of tasks, which is not desirable. One solution is to seed one common series on one task and to distribute the resulting pseudo-random numbers to all other tasks. This implies a load imbalance and requires additional MPI communication, yet it is for most pseudo-random number generators the only possibility. However, for the Mersenne Twister[3] (among others) Haramoto et al. (2008) found an efficient 'jump ahead' facility, i.e., a method to advance

the pseudo-random number generator state vector by $j = a \times 2^b$ steps ($a$ and $b$ integer with $a > 0$ and $b > 0$), without the need to harvest all $j$ pseudo-random numbers. Jumping ahead by numbers not representable in the form $a \times 2^b$ can be achieved by additionally harvesting $r$ pseudo-random numbers, such that $j' = a \times 2^b + r$. This procedure can nicely be used for a parallel decomposition independent method of parallel generation of pseudo-random number series and has been implemented in the MESSy infrastructure submodel RND. The same pseudo-random number series is seeded on all tasks, which then jump-ahead

and harvest independently, i.e. without additional communication overhead between the tasks. Each task can jump-ahead directly to the chunk of pseudo-random numbers it needs to harvest. The only prerequisite for this to work is that the number of required pseudo-random numbers per (each) task is a priori known to all other tasks. For instance, if for each ATTILA parcel $k$ pseudo-random numbers are required (e.g., per model time step), in total $k \times N$ pseudo-random numbers need to be harvested, i.e., $k \times n_i$ for task $i$. That means that task $i$ needs to jump-ahead by

$$j'_i = k \times \sum_{q=0}^{i-1} n_i \tag{2}$$

steps, before it can harvest its own chunk of $k \times n_i$ pseudo-random numbers.

    For the simulations analysed below, we used three uniformly distributed pseudo-random number series, all generated with the Mersenne Twister algorithm: for the boundary layer turbulence scheme, for the convection parametrisation, and for the initial placement of the Lagrangian parcels. The Monte Carlo diffusion was switched off.

---

[2]http://en.wikipedia.org/wiki/Marsaglia_polar_method
[3]only for uniformly distributed pseudo-random numbers, i.e., without the Marsaglia polar method



### 2.2.2 Advection

For every time step (in our simulations: $\Delta t = 600s$), the parcels are advected by the 3-dimensional wind field using a fourth order Runge-Kutta method. The wind field is interpolated on the parcel positions by linear interpolation horizontally (i.e., on the latitude - longitude grid) and by cubic Hermite interpolation vertically. The initialisation of the positions in the atmosphere is carried out randomly, so that the number of parcels corresponds to the mass of the respective model layer.

In the vertical direction we may use either $\eta$-coordinate vertical velocities ($\eta = \frac{p}{p_0}$, $\dot{\eta}$-kinematic velocity) calculated from the horizontal flux divergence using the continuity equation, or isentropic coordinates $\xi$ , where the vertical velocities $\dot{\xi}$ are calculated from the EMAC diabatic heating rates (diabatic velocity). The kinematic velocity is provided by default from EMAC, whereas the diabatic velocity was newly implemented similar to Eluszkiewicz et al. (2000) and Hoppe et al. (2016).

In our notation,

$$\xi = \theta\, f \tag{3}$$

with $\theta$ being the potential temperature and $f$ being defined as

$$\text{If} \qquad p > p_r \qquad f = sin\left(\frac{\pi}{2}\frac{1 - \frac{p}{p_s}}{1 - \frac{p_r}{p_0}}\right) \tag{4}$$

$$\text{If} \qquad p \le p_r \qquad f = 1 \;, \tag{5}$$

$$\tag{6}$$

with $\kappa = R_v/c_p p$ and $p$ being the atmospheric pressure. $p_r$ is the atmospheric pressure of the climatological tropopause, $c_p$ is the heat capacity at constant pressure. It characterises the transition from a pure $\theta$-coordinate system to the $\xi$-coordinate system. The standard surface pressure is $p_0 = 1013.25$ hPa, and $p_s$ is the actual surface pressure.

The vertical velocity is defined as

$$\dot{\xi} = \dot{\theta}\, f + \theta\dot{f} \;. \tag{7}$$

The diabatic vertical velocity in the troposphere $\dot{\xi}$ for $p > p_r$ appears as a mixed velocity between pure diabatic $\dot{\theta}$ and kinematic velocity in the troposphere according to Eq. 7. Only in the stratosphere $\dot{\xi}$ is a pure diabatic velocity.

### 2.2.3 Turbulence

Every parcel located within the planetary boundary layer (PBL) is randomly displaced in the vertical direction within the corresponding grid-cell. This stochastic mixing represents the boundary layer convective mixing process. The boundary layer height is calculated outside of ATTILA within the submodel TROPOP.

### 2.2.4 Convection

The LG convection scheme uses the mass-fluxes of the standard grid-box convection scheme in EMAC (submodel CON-VECT) to calculate the convective parcel movement. Therefore we will at first shortly introduce the convection scheme of





EMAC (Tiedtke, 1989; Nordeng, 1994), because the LG convection scheme bases on it. Convection in the standard convection scheme is initiated when convergence of moisture in a vertical column of the atmosphere exceeds a certain threshold value and a convectively unstable layer exists. Three types of convection are distinguished: Deep convection occurs, if moisture convergence through advection and evaporation at the surface takes place. Shallow convection, if moisture convergence is only

by evaporation at the surface, and mid-level convection, if the criteria of deep and shallow convection are not fulfilled but 90% relative humidity is reached within the planetary boundary layer.

Convection is parametrised by dividing a vertical column into an area of updraft (subscript $u$), downdraft (subscript $d$) and an area of compensating motion in the environment (subscript $e$). Convective transport in EMAC is parametrised only in vertical direction as a divergence of the tracer mass fluxes $F^u = M^u X^u$, $F^d = M^d X^d$, $F^e = M^e X^e$:

$$-\frac{1}{\overline{\rho}}\frac{\partial(\overline{\rho w' X'})}{\partial z}\bigg|_{conv} = -\frac{1}{\overline{\rho}}\left(\frac{\partial F^u}{\partial z} + \frac{\partial F^d}{\partial z} + \frac{\partial F^e}{\partial z}\right) \tag{8}$$

$\overline{X}$ is the tracer mass mixing ratio, $M$ is the mass flux, $\overline{\rho}$ is the air density. $\overline{w}$ is the vertical wind component, and $z$ the height. The quantities with an overbar are horizontal averages over the grid box, the quantities marked with a prime are the horizontal deviations from the respective grid box mean variables. $M^u \geq 0$ , $M^d \leq 0$ and $M^e$ are the mass fluxes of air for updraft, downdraft and the environment, respectively.

The change of mass fluxes with height is dependent on entrainment and detrainment fluxes:

$$\frac{\partial M^u}{\partial z} = E^u - D^u \tag{9}$$

$$\frac{\partial M^d}{\partial z} = E^d - D^d \tag{10}$$

with  $M^e = -(M^u + M^d) \tag{11}$

$E^u$ ($E^d$) comprise the entrainment (detrainment) rates due to turbulent exchange of mass through cloud edges and for the

updraft $E^u$ only, it implies the organised inflow associated with large-scale moisture convergence in cases of deep or mid-level convection. Accordingly, the detrainment rates include the turbulent exchange in up- and downdraft and, for the updraft only, the organised outflow at cloud top.

The corresponding tracer mass fluxes are

$$\frac{\partial F^u}{\partial z} = \frac{\partial M^u X^u}{\partial z} = E^u X^e - D^u X^u$$

$$\frac{\partial F^d}{\partial z} = \frac{\partial M^d X^d}{\partial z} = E^d X^e - D^d X^d$$

$$\frac{\partial F^e}{\partial z} = \frac{\partial(M^e X^e)}{\partial z}$$

The calculation of the convective transport of tracer mass starts with the determination of the type of convection (deep, shallow, mid-level). According to the estimated convective available potential energy (CAPE) the mass flux at cloud base is calculated. Further details of the calculation of the mass fluxes are described by Tiedtke (1989) and Nordeng (1994).

In our LG convection scheme air parcels can follow the updraft, downdraft or the compensating motion in the environment at a grid column with convection within one time step. The forcing used for the Lagrangian convection scheme is provided





by the mass-fluxes $M$ of the convection scheme of EMAC for updraft and downdraft, respectively. Probabilities $P$ for each level are calculated from the mass fluxes within a vertical column. Each LG parcel is equipped with a (pre-calculated) random number (see Section 2.2.1). For each parcel ascend (or descend) in an updraft (downdraft) is applied with probability $P$. The probability for an air parcel to follow the updraft $P$ is equal to the ratio of the mass of the air parcel moving into the updraft to

the mass of air at that level.

If $(M_k - M_{k+1}) > 0$, which means that the mass flux increases with height then

$$P_k = \frac{m_e}{m_g} = \frac{(M_k - M_{k+1})\,\Delta t\,g}{p_{k+1} - p_k} \tag{12}$$

with

$$m_g = \frac{(p_{k+1} - p_k)\,A}{g} \text{ and } m_e = (M_k - M_{k+1})\,A\,\Delta t \tag{13}$$

and $p$ - pressure (in hPa), $A$ - area (in m²), $M$ - air mass fluxes (in $\mathrm{kgm^{-2}s^{-1}}$), $g$ - gravity acceleration, $\Delta t$ - time step length (in s).

If $(M_k - M_{k+1}) < 0$, i.e, the mass flux decreases with height, a negative probability is defined to reflect a situation where a parcel may leave the updraft due to detrainment. The probability is equal to the ratio of the mass leaving the level to the mass entering the same level from below:

$P = (M_k - M_{k+1})/M_{k+1}$ . $\tag{14}$

The equations of the probability functions are analogous for the downdraft.

The LG convection scheme is strictly mass conserving. Thus for every time step the number of parcels per grid box after convection equals the number before convection (see n = const. in Fig.1), because every updraft and downdraft forces a compensating large-scale motion of parcels. The probability $P$ for subsidence is not estimated from the mass-fluxes provided

by EMAC. It is calculated for every layer, depending on the number of parcels that need to subside in order to fulfill the mass conservation for every layer.

## 2.3   Submodel LGGP: Transformation between Lagrangian and Eulerian representation

The submodel LGGP (LaGrangian to Grid Point transformations) performs the transformation of variables from Lagrangian representation to grid-point representation or vice versa. The variables (channel objects) to be transformed are specified by the

user in the &CPL-namelist of the submodel.

Transformations of a variable from LG to GP use the information of all parcels in the corresponding grid box and calculate either

- the sum of this variable over all parcels,

- the average of the variable over all parcels,

- the standard deviation of the variable over all parcels, or



– the average of the variable over all parcels, in which the variable is $> 0$.

Grid-boxes without parcels are either filled with a constant value (defined by the user in the &CPL-namelist) or with the value from a selected grid-point variable (defined as channel object in the &CPL-namelist).

The transformation from GP to LG distributes the variable onto all parcels in the respective grid box, either mass conserving (i.e. with equal share) or uniformly (i.e. with the same value of the GP variable). An example &CPL-namelist is shown in the supplement.

### 2.4 Submodel LGTMIX: Mixing of compounds in Lagrangian representation

The submodel LGTMIX (LaGrangian Tracer MIXing) calculates the exchange of tracer mass between Lagrangian parcels. Each Lagrangian parcel is described by a mathematical point. Its tracer mixing ratio represents a mean over the whole parcel.

Turbulence in the ambient air lead to a mixing of air of adjacent parcels. In order to avoid a parcel to parcel communication, we define a background mixing ratio $\bar{c}$, with which the parcel can communicate. The background is defined by the mean mixing ratio of the individual parcels $c_i$ within one grid box of the EMAC grid:

$$\bar{c} = \frac{1}{n} \sum_{i=1}^{n} c_i \ .$$ (15)

The altered mixing ratio of the respective parcel is then calculated by $c_i^{new} = c_i + (\bar{c} - c_i)\, d$, with $d$ being a dimensionless

mixing parameter within the range [0,1], which controls the magnitude of the exchange. The user can specify in the LGTMIX &CPL-namelist the mixing parameter $d$ individually for vertical model level ranges defined by two external layer definitions (i.e., external channel objects), such as the boundary layer height (from TROPOP), the tropopause (from TROPOP), or any surface provided by VISO (a diagnostic submodel to diagnose vertically layered 2-d iso-surfaces in 3-d scalar fields and to map 3-d scalar fields in GP representation on iso-surfaces (Jöckel et al., 2010)). The value for each of these layers can either

be a constant or a function defined by scaling an external grid-point variable (channel object) in a given range ($[min, max]$, to be specified by the user) to the interval $[0, 1]$. The $d$ in each of these layers can be scaled further for each tracer individually. An example &CPL-namelist is shown in the supplement. For our simulations discussed below, standard values of $d = 10^{-3}$ for the troposphere and $d = 5 \cdot 10^{-4}$ for the stratosphere, respectively, were selected similar as by Reithmeier and Sausen (2002). Additional tracers for diagnostic purposes have been simulated without mixing in the stratosphere (i.e., $d$ scaled by 0.0) and

with doubled mixing strength (i.e., $d$ scaled by 2.0) in the stratosphere, respectively.

### 2.5 Submodel LGVFLUX: Diagnostic of vertical fluxes through horizontal surfaces

The submodel LGVFLUX is a useful tool to calculate on-line vertical mass-fluxes through horizontal surfaces. Mass fluxes through a two-dimensional surface (e.g., isentropic surface, potential vorticity iso-surface, pressure level), are calculated by analysing the movement of LG particles through these surfaces (up- or downward) and summing over all particles which cross

the surface per unit time and area:

$$F_{sfc}(c) = \frac{\sum m_i \times c_i}{\Delta t\, A} \ ,$$ (16)





with $F_{sfc}$ being the mass-flux through the horizontal surface (indicated by the subscript $sfc$), and $m_i$ the mass of a LG parcel that is transported through the surface with area $A$ in time $\Delta t$ (i.e., the model time step length). For air, the mixing ratio $c_i$ is 1.0, for tracer mass fluxes $c_i$ is the corresponding tracer mixing ratio. In order to avoid summation over fast, reversible transitions, each surface definition is associated with a minimum residence time each parcel needs to reside after crossing the surface. For

5   taking into account this minimum residence time, each parcel is equipped with a clock to directly measure its transit time. If the parcel crosses a selected horizontal surface, its clock is started and will be reset only, if the parcel moves across the surface into the opposite direction. Thus, these "clocks" represent the transit time since passing through a specific surface. In Sect. 5 we present results calculated with this new tool to diagnose the stratosphere-troposphere exchange of air-mass and to estimate the AoA and the AoA spectra from the transit times in the stratosphere. An example &CPL-namelist is shown in the supplement.

## 2.6   DRADON

The submodel DRADON (diagnostic Radon tracer in GP space, (see Sect. 6.1 in Jöckel et al. (2010)) has been expanded to handle $^{222}Rn$ and its decay products also as tracers in Lagrangian representation (see Sect. 2.2.1). Here, we simulate $^{222}$Rn with a constant $^{222}$Rn source of 10000 atoms m$^{-2}$ s$^{-1}$ over ice-free land (zero elsewhere), and a decay with a half-life of 3.8 days as the only $^{222}$Rn sink.

15   Further, for the transformation of emission fluxes in GP space into Lagrangian tracer tendencies, the new routines of AT-TILA_TOOLS (see Sect. 2.2.1) have been used. As emission method we selected option 2 (see Sect. 2.2.1), i.e., we put the emitted $^{222}$Rn mass into those parcels, which reside lowest in the boundary layer. An example &CPL-namelist is shown in the supplement.

## 3   Observations

### 3.1   $^{222}$Rn

$^{222}$Rn has been frequently used for an evaluation of large-scale and convective transport processes (Dentener et al., 1999; Denning et al., 1998; Mahowald et al., 1995, 1997; Jacob et al., 1997; Gupta et al., 2004; Zhang et al., 2008; Jöckel et al., 2010), particularly due to its short life-time. We selected $^{222}$Rn measurements with an annual cycle at different sites over the globe as published by Zhang et al. (2008) and vertical profiles from Kritz and Rosner (1993) and Zaucker et al. (1996). $^{222}$Rn

25   is emitted from land surfaces due to a radioactive decay of radium in soils. $^{222}$Rn has a characteristically short radioactive half-life of ($\tau_{\frac{1}{2}} = 3.8$ days). For the evaluation of the simulated $^{222}$Rn distribution we use monthly mean surface values from 18 stations worldwide and selected vertical profiles. The large set of monthly mean surface values was collected from the literature by Zhang et al. (2008) and is used here for comparison. Observations of the vertical distribution of $^{222}$Rn are rare, especially if they cover more than the boundary layer. We use two different data sets of vertical profiles for comparison:





Kritz and Rosner (1993) used flights of the Kuiper Airborn Observatory in summer 1994 to achieve a representative selection of $^{222}$Rn measurements in the free troposphere. The flights were made from 3rd of June until the 16th of August 1994 around Moffett Field in California (37.4°N, 122°W), where 11 single profiles could be realised with a vertical resolution of 1 km.

Zaucker et al. (1996) compiled a data set from 9 flights in August 1993 from cities in Nova Scotia and New Brunswick on the east coast of Canada to the western North Atlantic Ocean during the North Atlantic Regional Experiment (NARE) Intensive. The vertical height of the measurements is restricted from the surface to about 5.5 km.

## 3.2 Age-of-Air

Mean age-of-air (AoA) is a common metric to quantify the overall capabilities of a global model to simulate stratospheric transport. It describes the transit time of air parcels in the stratosphere (Hall and Plumb, 1994). AoA is calculated (in model and observations) from an inert tracer with linearly increasing boundary conditions at the surface. AoA at a certain grid point in the stratosphere is then calculated as the time lag between the local tracer mixing ratio and the mixing ratio at a reference point (e.g., the boundary layer in the tropics). Inert tracers from observations are the anthropogenically emitted sulfur hexafluoride ($SF_6$) (Stiller et al., 2012; Haenel et al., 2015), and $CO_2$ (Engel et al., 2009; Andrews et al., 2001). Both will be used for comparison.

## 4 Model simulations

We performed two identical simulations with EMAC-ATTILA with respect to the climate, one uses the kinematic vertical velocity to drive the Lagrangian parcels, the other one the diabatic vertical velocity. The horizontal velocity remains equal in both simulations. EMAC was operated in T42L47MA resolution with 47 levels up to 0.01 hPa (see Sect. 2). We simulated the years 1950 to 2010 with prescribed sea surface temperatures (SSTs) from the global data set HadISST (available from http://www.metoffice.gov.uk/hadobs/hadisst/, Rayner et al. (2003)) similar as for the RC1-base-08 free-running simulation (see ESCiMo project description by Jöckel et al. (2016)). In contrast to the simulations within the ESCiMo project, due to limitations in working space, our simulations do not simulate interactive chemistry, however, monthly averages of radiatively active substances ($CO_2$, $O_3$, $N_2O$, $CF_2Cl_2$, $CFCl_3$) have been prescribed from RC1-base-08. Methane was initialised as in RC1-base-08 and the same pseudo emission time series (by Newtonian relaxation at the surface with the submodel TNUDGE) has been applied. However, methane oxidation and its contribution to stratospheric water vapour were treated in a simplified manner (with the submodel CH4): the oxidation educts OH, Cl and $O^1D$ have been prescribed as monthly averages from RC1-base-08. The photolysis rate $J(CH_4)$ was calculated with the submodel JVAL.

ATTILA was initialised with $1.15 \times 10^6$ parcels, which in sum represent the total mass of the atmosphere. The parcels were initially positioned according to the mass distribution in grid space. The results of the two simulations are further denoted as:

– GP for the results of the grid point simulation (EMAC); note that these are identical in both simulations.

– LG(diab) for the results of EMAC-ATTILA with diabatic vertical velocity.





- LG(kin) for the results of EMAC-ATTILA with kinematic vertical velocity.

The LG parcels are equipped with tracers with different properties:

- $^{222}$Rn is a commonly used tracer to study the vertical transport into the upper troposphere due to its characteristically short radioactive half-life of ($\tau_{\frac{1}{2}} = 3.8$ days). In our simulations it is emitted at the surface with an emission rate of 1.0 atom cm$^{-2}$ over ice-free land surfaces between 60°N and 60°S.

- SF$_6$_AoA and SF$_6$_AoAc are inert synthetic tracers. They differ with respect to the surface source. Both are nudged by Newtonian relaxation at the lowest model layer towards a linearly increasing mixing ratio. Note, for SF$_6$_AoA the linear in time increasing mixing ratio is latitude dependent. Using a spatially constant surface source (SF$_6$_AoAc) has the advantage, that concentrations differences in the atmosphere cannot have their origin in the distribution of surface sources.

- SF$_6$_AoA_nm has the same properties as the tracer SF$_6$_AoA, however, the inter-parcel mixing was set to zero (see 2.4). Hence, it can be used in comparison to SF$_6$_AoA to study the influence of local mixing between adjacent parcels on the global AoA distribution.

## 5 Evaluation

In the previous sections, we described a comprehensively updated version of the LG tracer transport scheme ATTILA, including a new LG convection scheme and the option to use a diabatic instead of the standard kinematic vertical velocity. In this section, we evaluate ATTILA by comparing the simulated $^{222}$Rn and SF$_6$_AoA tracer distributions with observations and with EMAC results, i.e., from the GP space.

### 5.1 Simulation of $^{222}$Rn

$^{222}$Rn has been frequently used for an evaluation of large-scale and convective transport processes (Dentener et al., 1999; Denning et al., 1998; Mahowald et al., 1995, 1997; Jacob et al., 1997; Gupta et al., 2004; Zhang et al., 2008; Jöckel et al., 2010). Jöckel et al. (2010) already showed, that the EMAC-model (version 2.40) is able to realistically simulate the $^{222}$Rn distribution. We therefore assume here, that our GP simulation with the EMAC-model (now version 2.53) simulates $^{222}$Rn similarly. A comparison with observations will follow in the next section. The inter-comparison of our new simulations of $^{222}$Rn between GP and LG space (Fig. 2) shows the largest values of $^{222}$Rn in the northern hemispheric boundary layer (the lowest 3 model layers). The large maxima north of 10°N and around 30°S are related to the surface emissions from of the large continents (Fig. 3). The small local maximum at 80° south is related to the surface edge of the Antarctic continent, where small land areas in the land sea mask generate local $^{222}$Rn emissions. In the boundary layer (north of 40°N) the zonal mean $^{222}$Rn values for LG(diab) are smaller than for LG(kin) and GP (Fig. 2). Contrary, south of 40°N the LG(diab) results are closer to the GP simulation. Part of the difference between the GP and the LG simulation is the uptake of emissions, that depend on the



number of LG parcels present in the lowest model levels (see also Sect. 2.6). The overall burden of $^{222}$Rn is the largest in the GP simulation. As expected, over the oceans (the remote regions) the $^{222}$Rn values are relatively small. At 100hPa in Fig. 2 the largest $^{222}$Rn values occur in the tropics. Here, GP and LG(diab) simulation results are in close agreement. LG(kin) simulates smaller values. The differences between LG(diab) and LG(kin) are related to the different vertical velocity scheme, because

this is the only difference between both LG schemes. In the higher latitudes the 100hPa level is already in the stratosphere and $^{222}$Rn has largely decayed due to its short half life time and the relatively long transport times in the stratosphere.

### 5.1.1   Annual cycle of $^{222}$Rn at the surface layer

We use $^{222}$Rn in-situ measurements at the surface layer of 18 stations distributed world-wide as they were published by Zhang et al. (2008). The model results are long-term averages of $^{222}$Rn in the surface layer over the years 1960-2000. They

are horizontally linearly interpolated to the respective location of the observations. Six stations (Crozet, Bermuda, Amsterdam Island, Kerguelen, Dumont and Mauna Loa) are far away from the continents and show the effect of long-range transport ($^{222}$Rn lower than 1000 Beq m$^{-2}$). Further six stations (Socorro, Cincinatti, Para, Puy de Dome, Beijing, and Hohenpeißenberg) are located on continents in the vicinity of the $^{222}$Rn sources. And finally six stations (Gosan, Hongkong, Cape Grim, Livermore, Bombay and Mace Head) are located at coastal sites influenced by the prevailing wind direction either from the sea or the

continent. A detailed comparison between model results (horizontal axis) and measurements (vertical axis) is shown in Fig. 4. Twelve monthly values of the mean annual cycle of GP, LG(diab) and LG(kin) – ∘ ⋆ △ – are presented as data points with different colours for each station. The results are similar to those of Jöckel et al. (2010, their Fig. 14). This cannot be taken for granted, because Jöckel et al. (2010) used a nudged simulation and a model setup with 90 vertical levels, whereas in our study the simulation is free running and our model setup has 47 vertical levels. Two stations are for all 12 monthly mean values

out of the thick-dashed area in Fig. 4: Beijing and Kerguelen (southern Indian ocean). Beijing's $^{222}$Rn values are too low and the Kerguelen values are too high simulated with all models. In Jöckel et al. (2010) the measured $^{222}$Rn values for Kerguelen are neither captured in the simulation. The simulated $^{222}$Rn values are strongly dependent on the local wind direction at the surface of the measurement sites, especially for the coast and the remote regions over sea.

### 5.1.2   Vertical profiles of $^{222}$Rn

We selected the vertical profiles of $^{222}$Rn from two campaigns: The NARE (North Atlantic Regional Experiment intensive; Zaucker et al. (1996)), and the MOFFET campaign at Moffet Field in California (Kritz and Rosner, 1993; Kritz et al. , 1998) from June to August 1994. The NARE campaign took place in the vicinity of Nova Scotia and Brunswick, Canada, in August 1993. Data were sampled over the ocean and over the continent. We used the simulation data of a climatological mean August (1960-2000) averaged over the region where the flights took place (60°W–70°W and 41°N–46°N). The triangles in Fig. 5

show the measured $^{222}$Rn in the atmosphere for several flights. The thick dashed curve is a spline interpolation on the grid of all flights. The $^{222}$Rn concentration decreases with height up to about 3 km and remains around $10^{-24}$ mol mol$^{-1}$. Both LG simulation results agree with the observations and are close to the GP results. The measurements of the MOFFET campaign (Fig. 6) shows a relatively large scatter of 11 single profiles in the free troposphere. The simulated $^{222}$Rn profiles in that region





were selected as a climatological mean for the months June to August (1960-2000) and capture the observations quite well. Observed $^{222}$Rn emissions, stemming from radioactive decay of radium in soils, are highly variable due to the dependence on the physical characteristics of the soil (Gupta et al., 2004). Therefore a certain spread between observations and model results is expected and acceptable.

## 5.2 Age-of-Air

The calculation of AoA is performed in two different ways: by a so-called clock-tracer (a linear in time increasing tracer like $SF_6$ or directly by a clock on a parcel (LG clock; Sect. 2.5). From a clock-tracer, AoA is calculated indirectly by comparing local tracer concentrations with reference concentrations, e.g. the surface concentration in the tropics (see Sect. 3.2). This concept is applied in the next section for the calculation of mean AoA. However, age spectra are calculated directly from transit times provided by the parcel clocks (e.g during the transit in the stratosphere). Conceptually, the calculated AoA differs, if it was calculated by a clock-tracer or by clocks. A clock-tracer is subject to inter-parcel mixing, which is not the case for parcel clocks. Therefore, the mean AoA calculated by the parcel clocks is older than for a clock-tracer. However, AoA from a simulated clock-tracer distribution can be directly compared to AoA from an observed tracer distribution.

### 5.2.1 Mean Age-of-Air

Mean AoA in the stratosphere is calculated from the $SF_6\_AoA$ tracer. The transit time is estimated by comparing the tracer mixing ratio in the stratosphere with the $SF_6\_AoA$ mixing ratio in the tropical boundary layer. Fig. 7 shows a comparison of $SF_6\_AoA$ at 20 km height between the GP, LG(kin) and LG(diab) results along with AoA derived from satellite observations from MIPAS (Stiller et al., 2012) and in-situ measurements of Waugh and Hall (2002) from $SF_6$ tracer, as well as from $CO_2$ in-situ data (Andrews et al., 2001). GP and LG(diab) show realistic distributions of AoA, although slightly lower AoA than the in-situ measurements. MIPAS data is known to overestimate AoA in the polar regions (Stiller et al., 2012). This is attributed to a known sink of $SF_6$ in the upper stratosphere, that is not accounted for in our simulations. Noticeably, the LG(diab) simulation is closer to the observations and LG(kin) shows up with a too low age. For the analysis, we therefore restrict our further evaluation to LG(diab). The mean age distribution (1960-2010) of LG(diab) is shown in Fig. 8. It confirms the well-known characteristics of the stratospheric Brewer-Dobson circulation with younger air in the tropical pipe and older air over the poles. Furthermore, the simulated AoA is slightly older in the whole stratosphere compared to GP (Fig. 9), most pronounced near the poles below 50 hPa. This difference is attributed to the Eulerian vertical velocity used in the flux-form semi-Lagrangian transport scheme for the GP simulations, that shows up- and downwelling at different high latitudes, which are not related to the net tracer transport (Hoppe et al., 2016).

### 5.2.2 Typical age spectra

AoA spectra are calculated directly from the clock transit times. These LG clocks represent the actual time a parcel resides in the stratosphere after it had crossed the tropopause level. However, these clocks do not "mix their time" with other parcels.





Therefore, the resulting spectrum might differ from age spectra calculated from so-called AoA "clock tracers" (Garny et al., 2013; Ploeger et al., 2014; Schoeberl et al., 2005). Typical age spectra for the tropics (20°N–20°S) and the poles (70°N–90°N, 70°S–90°S) between 50 hPa and 0.1 hPa are shown in Fig. 10. The frequency distribution of the LG(diab) simulation is calculated for every month for the years 1990-2010, binned into 0.5 year bins and normalised. We find the largest frequency
at a parcel age between 0 and 0.5 years in the tropics and an exponential shape of the spectrum. In contrast, the modal age is roughly 3 years at the poles. The shape of the spectrum is Gaussian with a positive skewness. The seasonal age spectra (selected between 400 K and 500 K and 50°N–70°N, see Fig. 11) show characteristic multiple local maxima along the time axis. The width of these maxima increase from MAM, JJA over SON to DJF. The distance of the maxima on the transit time axis is around 1 year. These maxima reflect the different contributions of air masses from the tropics and from high latitudes in the
seasonal cycle (Ploeger and Birner, 2016). The seasonal age spectra look qualitatively similar as in Fig. 5 of Ploeger and Birner (2016), although our modal values are about 0.3 years younger. The different modal values between Ploeger and Birner (2016) and our Fig. 11 are probably a consequence of the utilised concept in calculating the AoA. In LG(diab) we use our LG clocks to calculate the transit time in the stratosphere directly, if the parcels cross the tropopause level (see Section 2.5). Ploeger and Birner (2016) calculated their seasonal spectrum from an AoA clock tracer and the Green's Function (Waugh and Hall, 2002)
and relate their stratospheric concentrations to the tracer mixing ratio in the boundary layer to calculate the AoA. Therefore, the larger modal values in Ploeger and Birner (2016) refer to an additional transit time up to the tropopause level. The effect of these two different concepts on the calculation of AoA in the lower stratosphere is discussed further in Sect. 5.2.3.

### 5.2.3  Sensitivity of Age-of-Air to inter-parcel mixing

AoA is influenced by the amount of mixing between adjacent parcels. Inter-parcel mixing can be regarded as a diffusion
process leading to a reduction of local AoA gradients. The effect of inter-parcel mixing makes stratospheric air generally younger (Fig. 12). Stratospheric AoA without inter-parcel mixing as represented in LG(diab), described as LG(diabnm), is mostly up to 2.5 months older compared to the AoA with standard mixing (see Section 2.4), but slightly younger in the tropical lower stratosphere, where the mixing with upper tropospheric air becomes important. This "younger air through inter-parcel mixing" should not be mixed up with the "ageing by mixing" concept of Garny et al. (2013); Ploeger et al. (2014); Konopka
et al. (2015); Dietmüller et al. (2017). They calculate AoA from the tracer budget equation of AoA and distinguish between the different terms: the tendencies due to the residual stratospheric circulation, the tendencies of AoA due to eddy mixing and due to turbulent diffusion. Their concept allows to separate the contribution of mixing on the local AoA budget. In contrast, in our study we simply compare a simulated tracer with mixing to a tracer distribution without mixing (perturbation concept). Inter-parcel mixing in the troposphere (Fig. 13) has only a small and statistically insignificant effect on the simulated AoA,
because the troposphere is a well mixed region, where parcels often enough have contact with the surface source. The surface is the only region (in the vertical), where the tracer concentration for the tracer without inter-parcel mixing can be changed.





### 5.3 Stratosphere - troposphere exchange

The stratosphere-troposphere exchange (STE) is characterized by a global scale meridional circulation in which mass is transported upward in the tropics and downward in the extra-tropics (Holton et al., 1996). We use the new diagnostic submodel LGVFLUX to directly calculate the simulated mass flux through the tropopause in the LG simulation. The LG(diab) simulation captures these typical features (see Fig. 14 as an example for the mass flux through the 380 K isentropic surface) with a net upward flux in the tropics between $30°$S and $30°$N and a net downward flux from $40°$ to $90°$ north and south. Between $30°$and $40°$north and south the zonal mean net flux is near zero. Figure 15 shows the annual cycle of the net downward flux through the extra-tropical 380 K isentropic surface with a maximum in boreal winter for $30°$N-$90°$N and boreal summer for $30°$S-$90°$S. The annual amplitude between summer and winter is about $6 - 7 \times 10^9$ kgs$^{-1}$, and falls in the range given by Appenzeller et al. (1996) of $6 - 7 \times 10^9$ kg s$^{-1}$.

### 5.4 Lagrangian convection-statistics

The LG convective parcel movement depends on the calculated mass flux profile (from convection). We analysed the movement of parcels during deep convective events for the year 1997. The analysis of movement shows that within the updraft the largest number of parcels leave the boundary layer and are detrained into the free troposphere up to the tropopause (Fig. 16). The maximum levels of detrainment are between level index 43 and 38. Only a few parcels start to follow the updraft above the boundary layer (start level index < 45). Parcels in the downdraft (Fig. 17) start between levels 28 and 42 and most parcels are released into the boundary layer (level index 45 - 47). Interestingly, three height regions seem to preferably be the starting points for the downdraft: between level index 30 and 33, between 36 and 38 and at level 42. The compensating motion in the environment is a movement over a small distance only. The starting levels for subsidence comprise nearly the whole troposphere (Fig. 18). The most frequent movement is one level, but a few parcels are subject to a further downward movement (right side of the diagonal line). However, because the subsidence of parcels depends on a local probability (see 2.2.4), it is possible that even more parcels subside than originally should. This is then compensated in the next iteration by a rise of a parcel (left side of the diagonal line). This upward movement of parcels adds a certain amount of unphysical diffusion to the convection, that unfortunately cannot be avoided in this model setup.

## 6 Summary

In this study we described and evaluated the updated LG tracer transport scheme ATTILA. ATTILA was extended with a LG convection scheme and a formulation of diabatic vertical velocity. We implemented a submodel to describe inter-parcel mixing, so far set up with one parameter for the troposphere and one for the stratosphere, respectively. Moreover, the new submodel allows to implement more physically sound mixing parametrisations easily. New infrastructure submodels, which simplify the transformation between GP and LG space, the provision of random numbers in a parallel environment, and diagnostic submodels were developed. We performed 2 simulations from 1950 to 2010, both resulting in the same meteorological sequence



in GP. The simulations differ only with respect to the vertical velocity used for the LG model: one with a diabatic LG(diab) and one with the standard kinematic vertical velocity LG(kin). The annual cycle of the two LG simulations of $^{222}$Rn in the surface layer is in accordance with observation of a large number of stations and of comparable quality with a former nudged simulation in grid space. Vertical profiles of $^{222}$Rn measured during the NARE and the MOFFET agree within an acceptable

spread with our simulations. We expected the largest improvement of our results with respect to the simulation of AoA in the stratosphere in the LG(diab) simulation. Indeed, AoA in LG(diab) shows the best agreement with observations. Moreover, AoA spectra and the troposphere-stratosphere exchange are realistically simulated in LG(diab).

## 7 Data and Code availability

The Modular Earth Submodel System (MESSy) is continuously further developed and applied by a consortium of institutions.
The usage of MESSy and access to the source code is licensed to all affiliates of institutions which are members of the MESSy Consortium. Institutions can become a member of the MESSy Consortium by signing the MESSy Memorandum of Understanding. More information can be found on the MESSy Consortium Website (http://www.messy-interface.org). The code presented here has been based on MESSy version 2.53.0 and will be available in the next official release (version 2.55.0). The data from the simulations will be provided by the authors on request.

## 8 Authors contributions

SB and PJ did the implementation, performed the simulations, analysed the results and wrote the manuscript.

*Acknowledgements.* The data of the annual cycle of $^{222}$Rn at several locations in the surface layer was kindly provided by Kai Zhang. The data of the vertical Radon profiles are kindly provided by Holger Tost. The simulations were performed at the Leibniz-Rechenzentrum in Garching, Germany. The data was analysed with the interactive computer visualisation and analysis environment ferret. This work was
funded by the Deutsche Forschungsgemeinschaft (DFG) within the projects LAWA (Lagrangesche Simulation des globalen atmospärischen Wasserkreislaufs) under the grant EG 40/24-1, the DFG Forschergruppe SHARP (Stratospheric Change and its Role for Climate Prediction) and by the DLR-Project KliSAW (Klimarelevanz von atmosphärischen Spurengase, Aerosolen und Wolken) and the Earth-System Modelling Project of the HGF. We thank Oliver Reitebuch for the internal review of an early version of the manuscript.





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



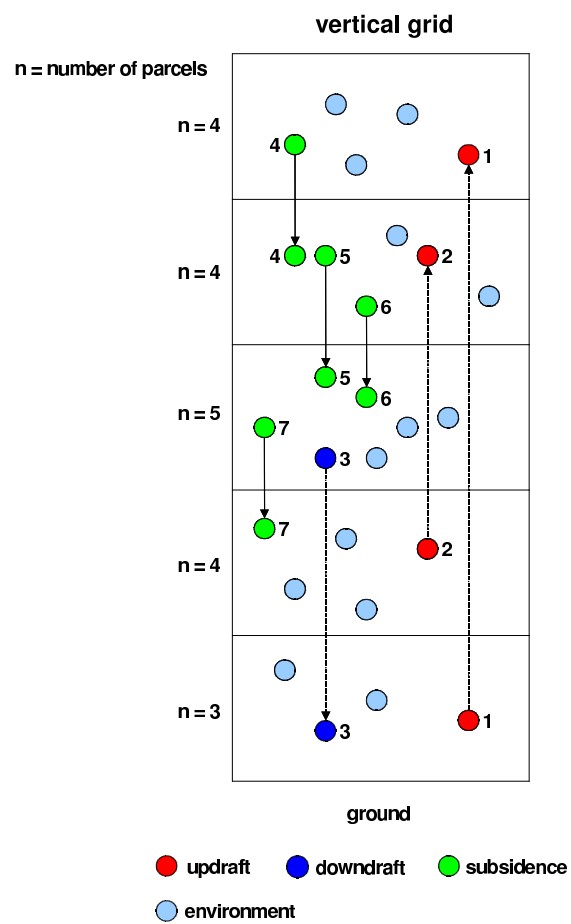

**Figure 1.** Mode of operation of Lagrangian convection in a vertical column. Coloured circles are Lagrangian parcels. n=3,4,5 is the original number of parcels in a grid box (chosen arbitrarily for this example), that should be reached again after the convective event to keep the air mass in each grid box constant.



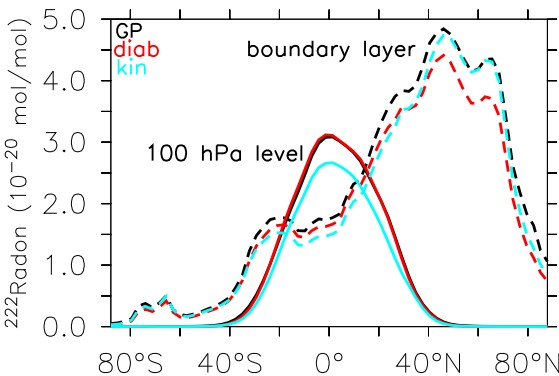

**Figure 2.** Zonal mean $^{222}$Rn concentration in the boundary layer (dashed) and at 100 hPa (solid, scaled with factor 10), GP - black line, LG(diab) - red line, and LG(kin) - light blue line.

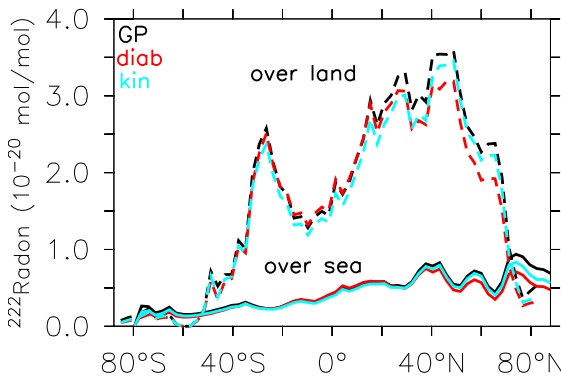

**Figure 3.** Zonal mean $^{222}$Rn concentration between 800 and 1013 hPa (weighted by the level thickness) over land (dashed) and over sea (solid) for GP (black line), LG(diab) (red line) and LG(kin) (light blue line).





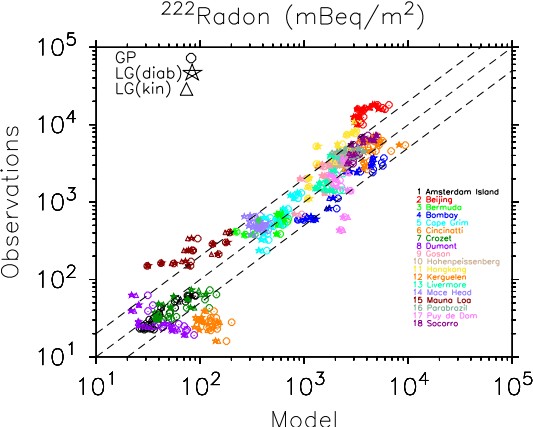

**Figure 4.** Monthly mean $^{222}$Rn [mBeq/m$^2$] surface concentrations: GP and LG model results against observations at different sites from Zhang et al. (2008). The thick dashed lines include a range within a factor of two of the observations.

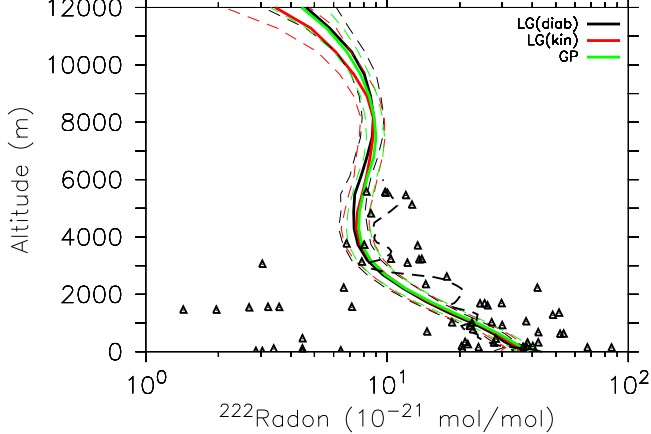

**Figure 5.** Vertical profiles of $^{222}$Rn mixing ratio [$10^{-21}$ mol/mol] during the NARE campaign. Dashed lines represent the one $\sigma$ standard deviation of the simulations. The thick dashed line is a spline interpolation of the scattered data on the grid.





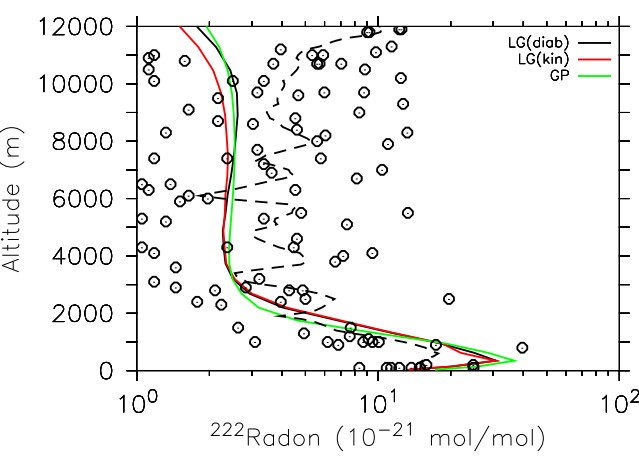

**Figure 6.** Vertical profiles of $^{222}$Rn mixing ratio $[10^{-21}\,\mathrm{mol/mol}]$ during the MOFFET campaign. The thick dashed line is a spline interpolation of the scattered data on the grid.





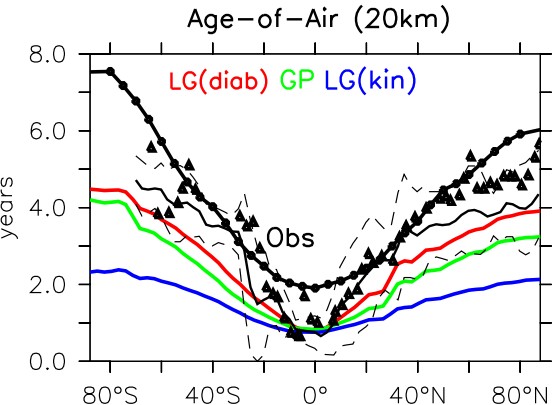

**Figure 7.** Zonal mean AoA at 20 km height (∼50hPa) from $SF_6$-AoA tracer (red: LG(diab), green: GP, blue: LG(kin), a mean over the years 2000-2010. Thick black line with circles: MIPAS data, a mean over the years 2003, 2007, 2008, 2009, 2010, 2011. Triangles: AoA from $SF_6$ at 20 km, from Waugh and Hall (2002). Thin black dashed lines: minimum and maximum AoA from $CO_2$ at 20 km from Andrews et al. (2001).

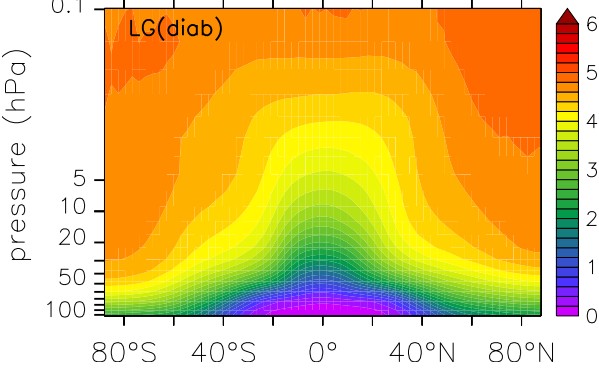

**Figure 8.** Zonal mean AoA (in years) from $SF_6$-AoA of the LG(diab) simulation.





**Figure 9.** Zonal mean difference of AoA (LG(diab)-GP) from $SF_6$-AoAc tracer (with constant source of $SF_6$ at the surface). NOT stippled area is statistically significant to the 99% level

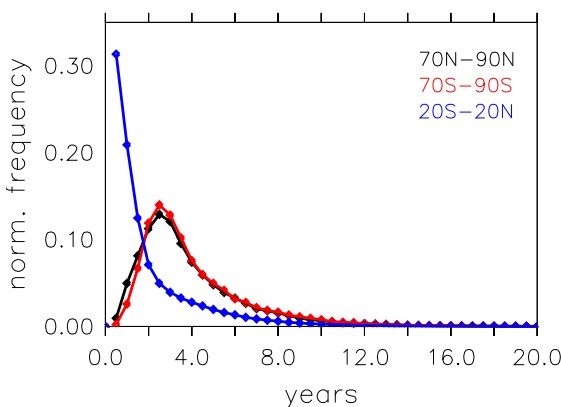

**Figure 10.** Normalised age spectra (tropics and poles) of the years 2006-2010 between 50hPa and 0.1hPa from the LG(diab) simulation. The data points represent the normalised frequency per half year bin.



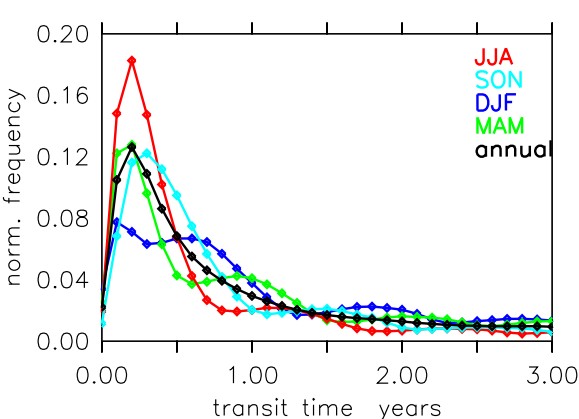

**Figure 11.** Normalised seasonal age spectrum from LG(diab) simulation of the years 1990-2010 between 400-500 K and 50°N-70°N.





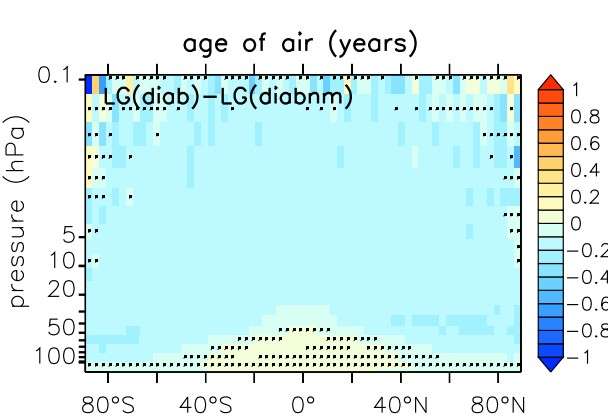

**Figure 12.** Zonal mean difference between and LG(diab) with standard mixing of $SF_6$-AoA tracer and LG(diab) with no-mixing (nm) between adjacent parcels. NOT stippled area is statistically significant to the 99% level).

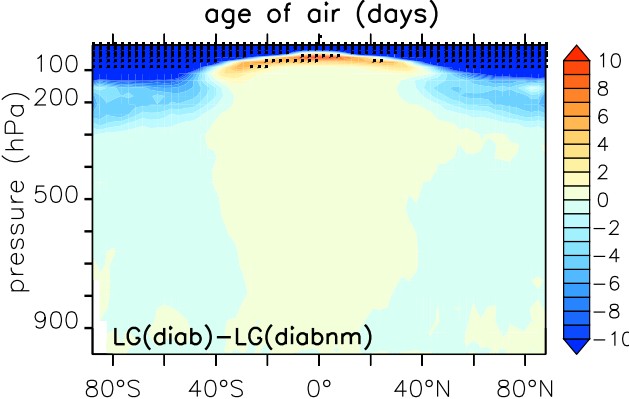

**Figure 13.** Zonal mean difference between LG(diab) and LG(diab-nm) with no inter-parcel mixing. Stippled area is statistically significant to the 99% level.

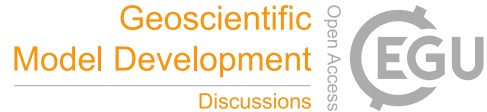


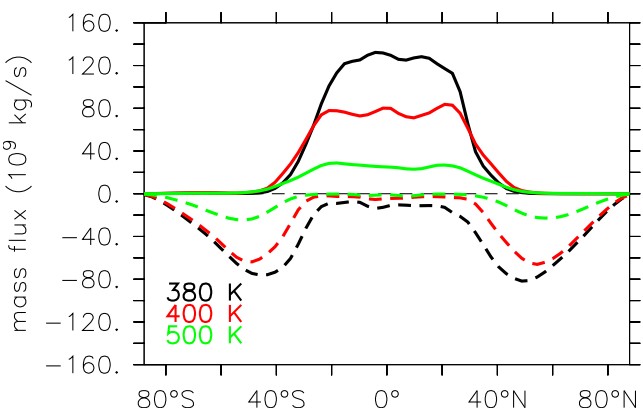

**Figure 14.** Zonal mean mass fluxes through the 380K (black), 400K (red), and 500K (green) isentropes. Upward (solid line) and downward (dashed line) from the LG(diab) simulation as a mean over (1960-2010).

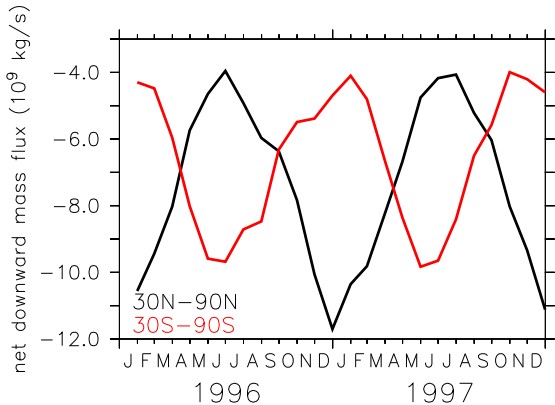

**Figure 15.** Monthly net mass fluxes (downward) from LG(diab) simulations through the 380 K isentrope for the northern and southern extra-tropics.





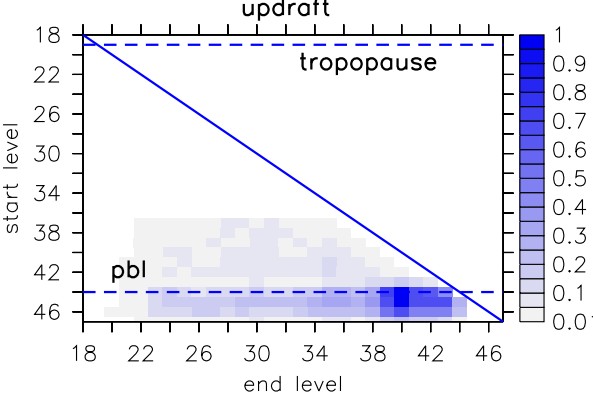

**Figure 16.** Movement characteristic of parcels, which are transported in the updraft in 1997. The vertical axis describes the start model level of a parcel. The horizontal axis describes the respective final updraft levels.

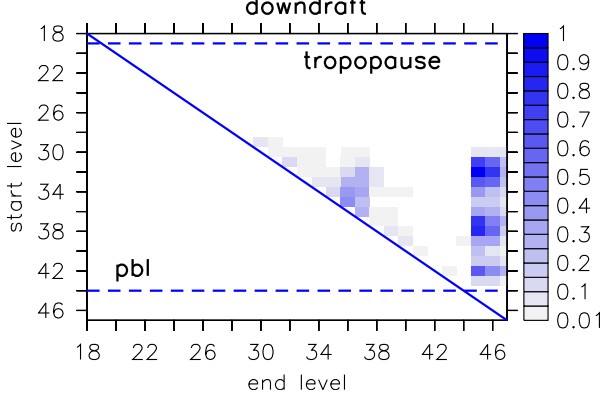

**Figure 17.** Movement characteristic of parcels, which are transported in the downdraft in 1997. The vertical axis describes the start model level of a parcel. The horizontal axis describes the respective final downdraft levels.





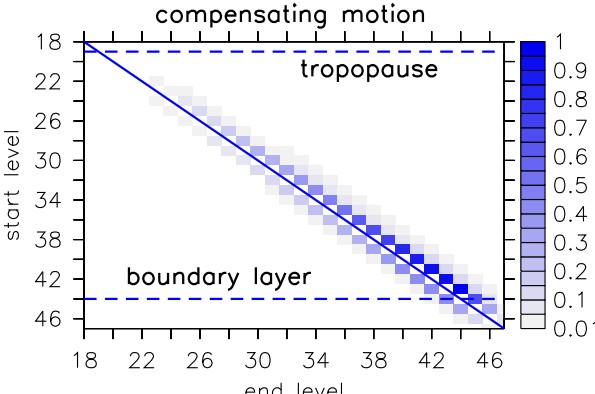

**Figure 18.** Movement characteristic of parcels, which represents the compensating movement in the environment (subsidence) in 1997. The vertical axis describes the start model level of a parcel. The horizontal axis describes the respective final subsidence levels. Note, that also a compensating upward parcel movement exist (shown below the blue diagonal line). For details see text.