# Peer review of "ATTILA 4.0: Lagrangian Advective and Convective Transport of Passive Tracers within the ECHAM5/MESSy (2.53.0) Chemistry Climate Model"

_Geoscientific Model Development, 2018_

## Referee Comment (RC1) · Anonymous Referee #1 · 23 Feb 2019

The study by Brinkop and Jöckel describes an extension of the Lagrangian transport module ATTILA, which is online coupled to the ECHAM5/Messy model through the Messy coupling framework. The extension includes several new modules for diabatic vertical velocities, convective transport, and inter-parcel mixing, as well as new tools for diagnosis and emission treatment. Furthermore, MPI paralellization (decomposing by parcels rather than by subdomains) and a careful treatment of the random number generator (essential for this type of modelling) have been implemented.

This reviewer considers these extensions as relevant and significant. The manuscript

is very well written and easy to follow. A model evaluation against Radon (troposphere) and Age-of-Air observations is presented, which highlights the benefits of diabatic vertical velocities to better represent the stratospheric age-of-air distribution. Although this is not a new finding, the model evaluation suggests that the implementation of diabatic velocities within ATTILA has been done properly.

The only weak point of the manuscript is the fact that the evaluation of the convective transport does not receive similar attention as the evaluation of the diabatic velocities. It would have been nice to see the effect of the convective transport on the vertical profiles of Radon in the troposphere. Why has this not been done? I clearly recommend publication of this manuscript with only minor corrections (and after addressing the weak point mentioned above).

Small points:

Introduction section: It would be useful to mention some typical (planned or past) applications of ATTILA End of page 4, beginning of page 5: In the list of new modules, it is not necessary to state "have been added" , "has been implemented" after each point. This could be stated once at the beginning, e.g. "the following new modules have been implemented:"

Page 5, Line 10: What do you mean by "transformations"?

Page 6, line 5: "depending whether" -> "depending on whether"

Page 8, line 21: I didn't really understand, how the "kinematic velocity" mixes with the "diabatic velocity" in equation 7. Rather it seems that vertical transport in these coordinates can occur by pressure changes (since f depends on pressure).

Page 10, Line 17: Isn't the convection scheme only mass conserving in the limit of a large number of air parcels? What happens if there is no air parcel available in the column that could be used to compensate the up- and downward motions?

Page 11, Line 14: Doesn't the mixing parameter d depend on the time step?

Page 13, Line 22: What do you mean by "working space"? Memory?

Page 15, Line 1: How can the overall burdens be different between the simulations, if the emissions of Radon are identical and Radon decays with a constant e-folding time?

Page 16 Line 3: You may also refer to Karstens et al. (2015): https://www.atmos-chem-phys.net/15/12845/2015/

Caption of Figure 12: "difference between and" -> "difference between"

---

## Referee Comment (RC2) · Anonymous Referee #2 · 24 Mar 2019

This paper described and evaluated the updated ATTILA (Atmospheric Tracer Transport in a LAgrangian model) coupled with the EMAC chemistry climate model. The model includes new physical routines for a Lagrangian convection scheme and a formulation of diabatic vertical velocity. New infrastructure submodels were also developed. The authors evaluated the results from grid point simulations (EMAC), EMAC-ATTILA simulations with diabatic vertical velocity and kinematic vertical velocity, respectively, against radon-222 surface and profile measurements. Their result shows an improvement of the tracer transport in the ATTILA with the diabatic (vs. kinematic) vertical

velocity. The documentation and evaluation are very useful, especially for their model users. Generally the paper is well written, but still requires more careful editing (see examples below). I recommend publication after minor revisions.

Minor comments:

P3, L19: "(ECHAM5, Roeckner et al., 2006)"

P3, L20 (and elsewhere): add comma after "i.e." (or "e.g.")

P3, L21: remove "MA-"

P4, footnote of Table 1: make it one single line.

P6, L7: "such as for instance" – remove "for instance".

P11, L23: "were selected similar as by Reithmeier and Sausen (2002)" – do you mean "following Reithmeier and Sausen (2002)"?

P12, L6: "only, if" — "only if"

P12, L12: remove "also".

P12, L13 and P14, L4-5: correct the unit on P14, and use the same unit.

P12, L24 and P13, L1: Kritz and Rosner (1993) was cited for the 1994 Radon profile data at Moffett Field. Should it be Kritz et al. (1998)?

P12, L26 and P14, L4: remove "of".

P13, L2: THE 3rd

P14, L9: "advantage that"

P14, L20-21: This is a repetition of what's said in the first 2 lines of section 3.1, and should be deleted. L22: "Jockel et al. (2010) showed that . . ."; L23: "assume here that. . .."; L26: "from the large"

P14, L27-28: "The small local maximum at 80 south is related to . . . where small land areas in the land sea mask generate local 222Rn emissions" – But it appears that Rn emissions in the model is only limited to 60S-60N (see top of page 14). Please clarify.

P15, L12 and Fig. 4 (panel & caption): "222Rn lower than 1000 Beq m-2)", "222Rn[mBeq/m2]" — the unit is incorrect. Please use "mBq/SCM" (i.e., mBq per standard cubic meter).

P15, L13: "And finally. . ." —- "Finally. . ."

P15, L16: remove the symbols

P15, L25-27: Again, these are repeating what's already said in section 3.1

P16, L2: remove "stemming from radioactive decay of radium in soils"

P16, L27: use "upwelling" instead of "up-" to avoid confusion.

P18, L15, Fig. 16: "The maximum levels of detrainment are between level index 43 and 38" — Are these shallow convection? Isn't it better to use altitude instead of model level index in the plot? What's the quantity shown on the color bar of Fig. 16-18?

P19, L4: during the . . .. campaigns

Fig.2: Are the concentrations averaged from the lowest 3 model layers? The caption needs this information. The concentrations at 100hPa are scaled up by a factor of 10, and it needs to be indicated on the panel, e.g., adding a right axis? Also explain what LG(diab) and LG(kin) are, or refer the reader to the text (section 4).

Fig.2-6: Consistently use mBq/SCM as the unit for 222Rn concentrations throughout the paper.

Fig. 5 caption: "Dashed lines" and "The thick dashed lines" are a bit confusing. "The thin dashed lines"? What are the triangles?

Fig.6 caption: what are the circles?

Fig.11: transit time (years)

Fig.12: typo "level)"

Fig.13: "Stippled area" or "NOT stippled area"?

Fig.14: The mass fluxes are plotted in "kg/s", which is dependent on the model grid-size (surface area). Without this model's grid-size information, other modelers cannot compare their results to this figure. Thus it's necessary to plot the mass fluxes in "kg/m2/s".

Fig. 15: "net downward mass flux" – remove "downward" since negative values already indicate "downward". Here it's ok to plot the mass fluxes in kg/s because the areas (30N-90N, 30S-90S) are given.

Suppl. Material: the cover page should use the same title as the one for the main text, and add one paragraph explaining what's included in the supplementary materials.

---

## Author Comment (AC1) · 15 Apr 2019

Please find our reply in the attached file.

Please also note the supplement to this comment:
https://www.geosci-model-dev-discuss.net/gmd-2018-302/gmd-2018-302-AC1-supplement.pdf

---

## Author Response (AR1)

**Reply to Referee 1**

*The study by Brinkop and Jöckel describes an extension of the Lagrangian transport module ATTILA, which is online cou­pled to the ECHAM5/Messy model through the Messy coupling framework. The extension includes several new modules for diabatic vertical velocities, convective transport, and inter-parcel mixing, as well as new tools for diagnosis and emission treatment. Furthermore, MPI parallelization (decomposing by parcels rather than by sub domains) and a careful treatment of the random number generator (essential for this type of modeling) have been implemented. This reviewer considers these extensions as relevant and significant. The manuscript is very well written and easy to follow. A model evaluation against Radon (troposphere) and Age-of-Air observations is presented, which highlights the benefits of diabatic vertical velocities to better represent the stratospheric age-of-air distribution. Although this is not a new finding, the model evaluation suggests that the implementation of diabatic velocities within ATTILA has been done properly.*

Reply: We thank the referee for these positive comments.

*The only weak point of the manuscript is the fact that the evaluation of the convective transport does not receive similar attention as the evaluation of the diabatic velocities. It would have been nice to see the effect of the convective transport on the vertical profiles of Radon in the troposphere. Why has this not been done? I clearly recommend publication of this manuscript with only minor corrections (and after addressing the weak point mentioned above).*

Reply:
Indeed, it seems like we did not pay too much attention to the effect of convection on the distribution of trace species. But this was not the case!
We performed different test simulations, though at a lower resolution (T21L19), but decided not to show them as part of the manuscript, because the findings do not help to evaluate the model. It is well known that convective tracer transport in the troposphere is essential to reproduce the vertical gradients. Thus, without the LG convection scheme we can hardly expect any meaningful results, which further cannot be evaluated (since there are no observations without convection).
Nevertheless, the results for Radon with and without LG convection are presented in Figure 1 below. The underlying simu­lation was a perpetual January simulation in T21L19 vertical resolution of EMAC (24 months).
Displayed are zonally averaged tropical vertical profiles of Radon simulated with ATTILA with and without LG convection in comparison to the grid-point calculation (which has been evaluated already earlier, see Jöckel et al., 2010).
For the reasons given above, we are hesitating to include such an analysis into the manuscript, because we think it is not of value.

*Small points:*
*Introduction section: It would be useful to mention some typical (planned or past) applications of ATTILA. End of page 4, beginning of page 5: In the list of new modules, it is not necessary to state "have been added" , "has been implemented" after each point. This could be stated once at the beginning, e.g. "the following new modules have been implemented:"*

Reply:
We introduced a small paragraph in the introduction, in which we describe, how ATTILA was used in the past, and describe future applications in the summary (now Summary and Outlook) section.

Further, we removed the "has been implemented" etc. as suggested.

*Page 5, Line 10: What do you mean by "transformations"?*

Reply:
Here, transformations describe, for instance, the conversion of variables between grid-point representation and Lagrangian representation, and vice versa.

We clarify this in the revised manuscript.

*Page 6, line 5: "depending whether" -> "depending on whether"*

Reply: Corrected.

*Page 8, line 21: I didn't really understand, how the "kinematic velocity" mixes with the "diabatic velocity" in equation 7. Rather it seems that vertical transport in these coordinates can occur by pressure changes (since f depends on pressure).*

Reply:
Equation (7) is simply the time derivative of equation (3). We add some text to clarify this.

*Page 10, Line 17: Isn't the convection scheme only mass conserving in the limit of a large number of air parcels? What happens if there is no air parcel available in the column that could be used to compensate the up- and downward motions?*

Reply:
This case cannot happen by design. If there would be indeed the case of only one parcel in a column, this single parcel would compensate itself and would therefore result in a vanishing net transport. Therefore, the LG convection scheme is even strictly conserving the local mass. To clarify this, we modified the text slightly.

*Page 11, Line 14: Doesn't the mixing parameter d depend on the time step?*

Reply:
In our specific setup the mixing parameter was set constant, i.e. independent of the time step, but differently for troposphere and stratosphere (as mentioned on page 11, lines 22-25). However, as outlined in line 20 of the same page, it can also be recalculated in every time step, depending on a time-dependent variable (channel object). We clarify this in the revised text.

*Page 13, Line 22: What do you mean by "working space"? Memory?*

Reply:
Indeed, we meant neither nor but more general computational resources. And since it is not really an argument (because simulations including chemistry would be possible, though computationally expensive) we removed the statement in the revision.

*Page 15, Line 1: How can the overall burdens be different between the simulations, if the emissions of Radon are identical and Radon decays with a constant e-folding time?*

Reply:
The differences occur, because the horizontal distribution of parcels in the boundary layer (where the source is picked up) differs slightly, depending on the chosen vertical velocity. And since Radon emissions occur only over land, differences in the burden cannot be excluded.
We clarify this in the revision and remove the sentence about the burden, because it is not discussed any further.

*Page 16 Line 3: You may also refer to Karstens et al. (2015): https://www.atmos-chem- phys.net/15/12845/2015/*

Reply: Thank you for the hint. We added the reference in the text.

*Caption of Figure 12: "difference between and" -> "difference between"*

Reply: Corrected.

[Figure]

**Figure 1.** Zonally averaged tropical Radon mixing ratio ($10^{19}$ mol/mol) versus pressure altitude from a perpetual January simulation with EMAC-ATTILA in T21L19 resolution. Black line: GP simulation, red line: Lagrangian simulation WITH LG convection, green line: LG simulation WITHOUT LG convection.

**Reply to Referee 2**

*This paper described and evaluated the updated ATTILA (Atmospheric Tracer Trans- port in a LAgrangian model) coupled with the EMAC chemistry climate model. The model includes new physical routines for a Lagrangian convection scheme and a formulation of diabatic vertical velocity. New infrastructure submodels were also developed. The authors evaluated the results from grid point simulations (EMAC), EMAC-ATTILA simulations with diabatic vertical velocity and kinematic vertical velocity, respectively, against radon-222 surface and profile measurements. Their result shows an improvement of the tracer transport in the ATTILA with the diabatic (vs. kinematic) vertical velocity. The documentation and evaluation are very useful, especially for their model users. Generally the paper is well written, but still requires more careful editing (see examples below). I recommend publication after minor revisions. .*

Reply: We thank the referee for these positive comments.

*Minor comments:*

*P3, L19: "(ECHAM5, Roeckner et al., 2006)"*

Reply: Done.

*P3, L20 (and elsewhere): add comma after "i.e." (or "e.g.")*

Reply: Comma added.

*P3, L21: remove "MA-"*

Reply: Done.

*P4, footnote of Table 1: make it one single line.*

Reply: Done.

*P6, L7: "such as for instance" – remove "for instance".*

Reply: Done.

*P11, L23: "were selected similar as by Reithmeier and Sausen (2002)" – do you mean "following Reithmeier and Sausen (2002)"?*

Reply: Yes, we meant "following Reithmeier and Sausen (2002)" and corrected it in the revised manuscript.

*P12, L6: "only, if" — "only if"*

Reply: Comma removed.

*P12, L12: remove "also".*

Reply: Removed.

*P12, L13 and P14, L4-5: correct the unit on P14, and use the same unit.*

Reply: Corrected.

*P12, L24 and P13, L1: Kritz and Rosner (1993) was cited for the 1994 Radon profile data at Moffett Field. Should it be Kritz et al. (1998)?*

Reply: The referee is right. We corrected this in the revised manuscript.

*P12, L26 and P14, L4: remove "of".*

Reply: Corrected.

*P13, L2: THE 3rd*

Reply: "THE" was added.

*P14, L9: "advantage that"*

Reply: Corrected.

*P14, L20-21: This is a repetition of what's said in the first 2 lines of section 3.1, and should be deleted. L22: "Jöckel et al. (2010) showed that . . ."; L23: "assume here that. . ..."; L26: "from the large"*

Reply: We deleted the sentence of lines 20/21 and corrected the following text passage.

*P14, L27-28: "The small local maximum at 80 south is related to . . . where small land areas in the land sea mask generate local 222Rn emissions" – But it appears that Rn emissions in the model is only limited to 60S-60N (see top of page 14). Please clarify.*

Reply: Thank you for this hint. Indeed, the emissions are over 90S-90N and not restricted to 60N-60S. We corrected the manuscript.

*P15, L12 and Fig. 4 (panel and caption): 222Rn lower than 1000 Beq m-2), "222Rn[mBeq/m2]" - the unit is incorrect. Please use "mBq/SCM" (i.e., mBq per standard cubic meter).*

Reply: We corrected the units.

*P15, L13: "And finally. . ." —- "Finally. . ."*

Reply: Corrected.

*P15, L16: remove the symbols*

Reply: Done.

*P15, L25-27: Again, these are repeating what's already said in section 3.1*

Reply: We removed the sentence from the manuscript.

*P16, L2: remove "stemming from radioactive decay of radium in soils"*

Reply: Done.

*P16, L27: use "upwelling" instead of "up-" to avoid confusion.*

Reply: Corrected.

*P18, L15, Fig. 16: "The maximum levels of detrainment are between level index 43 and 38" — Are these shallow convection? Isn't it better to use altitude instead of model level index in the plot? What's the quantity shown on the color bar of Fig. 16-18?*

Reply:
Only deep convective events are considered for this figure. However, your question triggered us to control our calculation with respect to the maximum level of detrainment. And in fact, we found out that a wrong data file was selected for the Figures 16-18. We corrected this and now provide correct figures. Additionally, we calculated the start and end levels in pressure levels instead of level numbers. Fig. 16 now shows the maximum detrainment level between 500 and 600 hPa. The color bar displays the number of moving parcels from a start level to its respective end level in the tropics between 20° north and south, normalised by the maximum number. The caption is modified in the revised manuscript.

*P19, L4: during the . . .. campaigns Fig.2: Are the concentrations averaged from the lowest 3 model layers? The caption needs this information. The concentrations at 100hPa are scaled up by a factor of 10, and it needs to be indicated on the panel, e.g., adding a right axis? Also explain what LG(diab) and LG(kin) are, or refer the reader to the text (section 4).*

Reply:
The word "campaigns" is added to the revised manuscript. Yes, the concentrations are averaged over the lowest 3 model layers. We added this information to the caption. We added a right axis for the (lower) concentrations at 100 hPa in Fig.2 and put the information on the LG(diab) and LG(kin) simulations into the caption.

*Fig.2-6: Consistently use mBq/SCM as the unit for 222Rn concentrations throughout the paper.*

Reply: We modified the units consistently throughout the manuscript.

*Fig. 5 caption: "Dashed lines" and "The thick dashed lines" are a bit confusing. "The thin dashed lines"? What are the triangles?*

Reply: Yes, the text is confusing. We reformulated the caption.

*Fig.6 caption: what are the circles?*

Reply: We added the explanation to the caption.

*Fig.11: transit time (years)*

Reply: Corrected.

*Fig.12: typo "level)"*

Reply: Done.

*Fig.13: "Stippled area" or "NOT stippled area"?*

5 Reply: We meant "Stippled area" as stated in the caption.

*Fig.14: The mass fluxes are plotted in "kg/s", which is dependent on the model grid- size (surface area). Without this model's grid-size information, other modelers cannot compare their results to this figure. Thus it's necessary to plot the mass fluxes in "kg/m2/s".*

Reply: We have modified the figure. The mass fluxes are now in kg/m2/s.

*Fig. 15: "net downward mass flux" – remove "downward" since negative values already indicate "downward". Here it's OK to plot the mass fluxes in kg/s because the areas (30N-90N, 30S-90S) are given.*

Reply: Done.

*Suppl. Material: the cover page should use the same title as the one for the main text, and add one paragraph explaining what's included in the supplementary materials.*

Reply:
We changed the title and included a short paragraph describing the presented material in the supplement.

[revised manuscript text omitted]